# Recent Research on Combination of Radiotherapy with Targeted Therapy or Immunotherapy in Head and Neck Squamous Cell Carcinoma: A Review for Radiation Oncologists

**DOI:** 10.3390/cancers13225716

**Published:** 2021-11-15

**Authors:** Daniel Tao Xing, Richard Khor, Hui Gan, Morikatsu Wada, Tai Ermongkonchai, Sweet Ping Ng

**Affiliations:** 1Oliver Newton-John Cancer Wellness and Research Centre, Austin Health, Heidelberg, VIC 3079, Australia; daniel.xing@austin.org.au (D.T.X.); richard.khor@austin.org.au (R.K.); morikatsu.wada@austin.org.au (M.W.); 2Medical Oncology, Austin Hospital, Heidelberg, VIC 3079, Australia; hui.gan@austin.org.au; 3La Trobe University School of Cancer Medicine, Heidelberg, VIC 3079, Australia; 4Department of Medicine, University of Melbourne, Melbourne, VIC 3000, Australia; 5Faculty of Medicine, Dentistry and Health Science, University of Melbourne, Melbourne, VIC 3000, Australia; permongkonch@student.unimelb.edu.au

**Keywords:** radiotherapy, epidermal growth factor receptor, immunotherapy, cetuximab, stereotactic body radiotherapy, abscopal effect

## Abstract

**Simple Summary:**

Radiotherapy (RT) is an effective treatment for head and neck squamous cell carcinoma (HNSCC). Concurrent RT with high dose cisplatin (100 mg/m^2^, days 1, 22, and 42) is the standard of care (SOC) for non-operative HNSCC in curative settings, however, it is associated with both significant toxicities. In this review, we discussed the evidence of combination of anti-epidermal growth factor receptor, cetuximab, or immune checkpoint inhibitors (ICIs) with RT to compare with SOC. Cetuximab has been shown to be a less effective agent than cisplatin multiple recent trials, but it remains a reasonable alternative for those who are not fit for cisplatin. ICIs are active agents in recurrent and metastatic HNSCC. The role of ICIs with RT in the curative setting is yet to be defined. Multiple clinical trials are currently recruiting. Combining ICIs with stereotactic body radiotherapy (SBRT) is an attractive treatment in patients with oligometastatic or oligoprogressive HNSCC to boost the anti-tumor immune response.

**Abstract:**

Radiotherapy plays an important role of managing head and neck squamous cell carcinoma (HNSCC). Concurrent radiotherapy with radiosensitizing cisplastin chemotherapy is the standard of care (SOC) for non-operable locally advanced HNSCC. Cetuximab, a monoclonal antibody of epidermal growth factor receptor, was the most extensively studied targeted therapy as a chemo-sparing agent that was used concurrently with radiotherapy. Immunotherapy is used in the treatment of metastatic HNSCC. There is evidence to support the synergistic effect when combining radiotherapy with immunotherapy to potentiate anti-tumor immune response. There has been increasing interest to incorporate immune checkpoint inhibitor (ICI) with radiotherapy in the curative setting for HNSCC. In this review, we discuss the latest evidence that supports concurrent radiotherapy with cisplatin which remains the SOC for locally advanced HNSCC (LA-HNSCC). Cetuximab is suitable for patients who are not fit for cisplatin. We then summarize the clinical trials that incorporate ICI with radiotherapy for LA-HNSCC in concurrent, neoadjuvant, and adjuvant settings. We also discuss the potential of combining immunotherapy with radiotherapy as a treatment de-escalating strategy in HPV-associated oropharyngeal carcinoma. Finally, the pre-clinical and clinical evidence of the abscopal effect when combining stereotactic body radiotherapy with ICIs is presented.

## 1. Introduction

Each year, 700,000 new cases of head and neck squamous cell cancer (HNSCC) are diagnosed worldwide [1]. For curative intent treatment, radiotherapy (RT) plays a crucial role in both early and advanced stage disease as a primary treatment modality to preserve swallowing and speech functions in locally advanced HNSCC. Concurrent radiosensitizing chemotherapy agent (i.e., cisplatin 100 mg/m^2^ administered every three weeks or 40 mg/m^2^ administered weekly) with radiotherapy is the current standard of care in locally advanced HNSCC for various head and neck sites [2]. While concurrent chemoradiation offers improved survival outcomes compared to radiotherapy alone, it is associated with acute and late toxicities including mucositis, myelosuppression, nausea/vomiting, hearing loss, and dysphagia that affects patients’ quality of life [3]. Search for equally effective but less toxic targeted therapy (e.g., anti-epidermal growth factor) with radiotherapy has been explored in both pre-clinical studies and clinical trials. 

It has been estimated that between 10 and 20% of patients are diagnosed with distant metastasis at presentation [4,5,6,7], and up to 40% of patients ultimately fail distantly after initial curative treatment for locoregional disease with multimodality treatment [8,9]. 

A combination of cetuximab, an epidermal growth factor receptor antibody, with cisplatin and 5-FU chemotherapy was the first-line systemic therapy of choice for recurrent or metastatic HNSCC [10]. Immune check-point inhibitors, including nivolumab and pembrolizumab, have also been approved by US Food and Drug Administration (FDA) and Australian Pharmaceutical Benefits Scheme (PBS) as second line systemic treatment [11], or first-line monotherapy in patients with high PD-L1 expression [12]. However, the overall response rate is modest even with immunotherapy, up to 30% at best. There is substantial evidence that local irradiation of the tumor can also stimulate systemic immune responses and lead to enhanced tumor cell recognition and killing by immune system [13,14] in various types of tumors including HNSCC. Current trials are investigating the combination of radiotherapy with these agents in both curative and metastatic settings [15,16] in HNSCC. The combination of immunotherapy with radiotherapy may increase the ability to induce immunogenic death.

Recent studies have demonstrated that HNSCC can be divided into two distinct subgroups based on Human Papilloma Virus (HPV) status. While the ‘traditional’ p16/HPV- negative HNSCC is associated with environmental exposures to carcinogens such as tobacco and alcohol, p16/HPV-positive HNSCC is associated with HPV infection with no traditional risk factors such as smoking. HPV-associated HNSCC has been established as a cause of oropharyngeal cancer [17,18], and data suggests that its incidence is rising [19,20] in western countries. Patients with HPV-associated oropharyngeal squamous cell carcinoma (HPV OPSCC) have substantially more favorable disease control rates and overall survival outcome [21,22,23] compared to their HPV-negative counterparts. Studies have focused on examining treatment de-escalation (with the intent of reducing treatment-related toxicities whilst preserving anti-tumor efficacy) in those with HPV OPSCC by combining radiotherapy with targeted therapy or immunotherapy to replace cytotoxic chemotherapy. 

In this review, we will describe the rationale of combining radiotherapy with epidermal growth factor receptor (EGFR) targeting and immunotherapy in locally advanced HNSCC (LA-HNSCC) or recurrent/metastatic HNSCC, together with clinical advances in novel treatment regimen. Nasopharyngeal carcinoma, which represent a distinct subgroup of head and neck cancers, often associated with Epstein–Barr virus infection, are not included in this overview.

## 2. Cetuximab and Radiation

The epidermal growth factor receptor (EGFR) is abnormally activated in epithelial cancers, including head and neck cancers. Overexpression of EGFR is an independent determinant of survival and predictor of locoregional relapse in patients with HNSCC [24,25]. Targeting EGFR is the most extensively studied targeted therapy in HNSCC to date. Cetuximab is a monoclonal EGFR antibody that has demonstrated the capacity to potentiate the effects of radiation in pre-clinical models [26]. It has been used as an alternative chemotherapy-sparing radiosensitizing agent (e.g., cisplatin) for patients with HNSCC [27] who are treated with curative intent. In a pivotal phase III trial, Bonner et al. demonstrated improved locoregional control (LRC, HR 0.68; *p* = 0.005), progression free survival (PFS, HR 0.70; *p* = 0.006), and overall survival (OS, HR 0.74; *p* = 0.03) when cetuximab was added to definitive RT compared to RT alone for locally advanced HNSCC (LA-HNSCC). Based on this study, cetuximab bioradiotherapy together with high-dose cisplatin-based concurrent chemoradiotherapy were both approved by FDA for treatment of head and neck cancer. In recurrent or metastatic HNSCC, adding cetuximab to a platinum/5-fluorouracil doublet (the EXTREME regimen) showed a significant improvement in OS and was the standard first-line palliative treatment [10]. The benefit of adding cetuximab is independent of HPV status in the post hoc-analysis, although the HPV-associated HNSCC had a better response to the EXTREME regimen in recurrent or metastatic settings [28]. 

With the aim to improve LRC and OS in locally advanced HNSCC, several studies have investigated treatment intensification by adding cetuximab to concurrent chemoradiotherapy (CT-RT) or adding induction chemotherapy to cetuximab bioradiotherapy (cetux-RT). The RTOG 0522 was designed to add cetuximab to the radiation-cisplatin platform, however, it failed to show the benefit of adding cetuximab to high dose cisplatin with radiotherapy in LA-HNSCC [29]. In contrast, the GORTEC 2007-01 trial did show an improved PFS and LRC by adding 3 cycles of carboplatin and fluorouracil to cetux-RT compared with cetux-RT alone [30], but there was no difference in OS. In the GORTEC 2007-02 trial, which was conducted at the same time, the addition of induction chemotherapy (taxotere, cisplatin, fluorouracil, or TPF) followed by cetxu-RT showed no difference in outcomes between the two arms [31]. The concurrent chemotherapy regimen was carboplatin and fluorouracil in both the GORTEC 2007-01 and GORTEC 2007-02 trials. 

However, historically, cetuximab was only compared to RT alone but limited comparison was performed with cisplatin in HNSCC. In the TREMPLIN trial [32], concurrent cetuximab with RT was compared with concurrent cisplatin-RT followed by TPF induction chemotherapy in laryngeal cancer. There was no evidence that one treatment was superior to the other, but the number of the patients was small and follow-up was short, which made the conclusion unconvincing [32]. An early retrospective trial from Memorial Sloan Kettering Cancer Centre showed that the cetux-RT was inferior to cisplatin-RT in OS and LRC [33], which casted some doubt on the efficacy of cetuximab compared to cisplatin. The direct comparison of concurrent cetuximab-RT to cisplatin-RT was recently conducted in a randomized trial HPV OPSCC with the study question of if cetuximab can be used as a treatment de-escalation agent as the Bonner’s trial did not show increased toxicity in the cetux-RT arm compared with RT alone [34]. In the high-risk population [21] (T1-T2, N2a-N3 M0 or T3-T4, N0-N3 as per AJCC 7), the NRG-RTOG 1016 [35] was designed as a non-inferior trial to compare concurrent cisplatin (100 mg/m^2^, days 1, 22, total 200 mg/m^2^) with cetuximab (400 mg/m^2^ loading dose then 250 mg/m^2^ weekly) with 70 Gy radiotherapy delivered over 6 weeks. In total, 805 eligible patients were enrolled. The non-inferiority for cetuximab was pre-specified if the 1-sided 95% upper confidence bound for the hazard ration (HR, cetuximab/cisplatin) is <1.45 (i.e., HR of 1.45 means that patients who were treated with cetuximab could have 1.45 times risk of death). Even with this generous non-inferiority margin the aim was not met. At 5 years follow up, the cisplatin arm had a better 5-year OS (85% vs. 78%, *p* = 0.02) and PFS (78% vs. 67%, *p* < 0.001) compared to the cetuximab arm. The cetuximab was considered not non-inferior to cisplatin. Although the worst grade for overall acute and late toxicities was no different between the two arms, those who received cetuximab had 40% more acute toxicity burden compared with those who had cisplatin. Similarly, in the low-risk HPV-associated oropharynx cancer population (non-smokers or lifetime smokers with a smoking history of <10 pack-years), the De-ESCALaTE trial, which randomized 334 patients to concurrent cetuximab or high dose cisplatin with 70 Gy radiotherapy, showed no difference in overall all-grade toxicity events per patient [36]. Although this study was not powered to show a difference in OS, it did demonstrate poorer OS at 2 years (97.5% vs. 89.4%, *p* = 0.001) and significantly worse LCR and distant control in the cetuximab arm compared to the cisplatin arm. In the subgroup analysis, by excluding patients with advanced T4 or N3 disease (i.e., stage I/II per AJCC 8th edition, very low risk population), concurrent cisplatin continues to demonstrate significantly better OS at 2 years (98.4% vs. 93.2, *p* = 0.043) compared to cetuximab. Further evidence for cisplatin was demonstrated in the TROG 12.01 trial [37] which showed that concurrent cetuximab did not reduce symptom severity as measured by the MD Anderson Symptom Inventory Head and Neck Symptom Severity Scale (MDASI-HN) and was associated with poorer failure free survival at 3 years (80% vs. 93%, *p* = 0.015). In summary, the recent studies demonstrate that cisplatin-RT remained the standard of care in HPV-associated oropharynx cancer. Cetuximab remained an option for those who are not fit for concurrent cisplatin chemotherapy.

## 3. Emergence of Immunotherapy in HNSCC

While it has been well documented that the development of traditional HNSCC is directly linked to carcinogens such as tobacco, alcohol, or HPV infection, there is increased recognition that defects in the immune response play major roles in the establishment and progression of these cancers [38]. The mechanisms include: (1) immune cell dysfunction within the tumor microenvironment (TME) and in the peripheral blood of patients with HNSCC [39]; (2) defects in antigen presenting of tumor cells [40]; (3) secretion of cytokines in the TME in favor of immunosuppression (e.g., TGF-beta) [41]; (4) presence of tumor immunosuppressive cells (e.g., T regulatory cells, tumor associated macrophages, and myeloid derived suppressor cells) [42]; and (5) upregulation of immune checkpoints, including PD-L1 and CTLA-4 [43]. A detailed review of the immune landscape in HNSCC is beyond the scope of this review. Until 2019, a combination of cetuximab/cisplatin/5-FU (EXTREME regimen) was the first-line systemic therapy of choice for recurrent or metastatic HNSCC [10]. The treatment paradigm is changing rapidly with the evidence that immunotherapy with checkpoint inhibitors is active in HNSCC. In 2016, both nivolumab and pembrolizumab received FDA approval for use in platinum-refractory patients with recurrent/metastatic HNSCC based on the CheckMate 141 [11] and Keynote-012 study findings [44]. Pembrolizumab is now FDA-approved as first-line monotherapy in patients with PD-L1 expression (CPS ≥ 1) and also in combination with platinum and 5-FU [12]. Multiple treatment strategies combining ICIs are under investigation, for example, nivolumab plus ipilimumab (CheckMate714 NCT02823574 and CheckMate651 NCT02741570) and durvalumab + tremelimumab (KESTREL NCT02551159). 

## 4. Radiation and Immune Response

There is evidence that locally applied radiation can also stimulate systemic immune responses, leading to enhanced tumor cell recognition and ultimately anti-cancer immunity [13]. Radiation not only causes lethal damage to the tumor cells to release tumor associated antigens (TAAs) [45], but also enhances MHC class I surface expression [46], calreticulin expression [47], and release of HMGB1, a damage-associated molecular pattern (DAMP). These events can lead to dendritic cell (DC) activation, an important arm of the robust immune system [48]. Radiation-induced cytokine release, principally type I and type II interferons, also play a role in DC recruitment [49]. The activated DCs migrate to lymph node to present the antigen to T cells [46] and result in tumor-specific T-cell activation and proliferation [50]. T-cell activation alone is insufficient for tumor eradication. In addition, radiation can encourage lymphocytes to infiltrate into the tumor by two main mechanisms: (1) normalizing tumor vasculature [51] and increasing the expression of endothelial adhesion molecules [52] to enhance immune-cell extravasation; (2) releasing chemokines to attract immune-cell migration and invasion [53]. The finer detail of the radiation-induced tumor immunity is extremely complex and yet to be fully understood, but there is an increasing body of evidence to support the combination of radiotherapy with immunotherapy [14,54].

## 5. Combining Radiotherapy with Immunotherapy in the Curative Setting

Since the success of using ICIs in the recurrent/metastatic HNSCC, current research focuses on incorporating ICIs into curative treatment. The rationales for the combination are: (1) using immunotherapy as an alternative to cetuximab for those who are unfit for cisplatin chemotherapy; (2) treatment escalation in patients with high-risk disease; (3) as neoadjuvant therapy to select the responders for treatment de-escalation; (4) as an alternative systemic agent from cisplatin for patients with favorable prognosis (e.g., HPV OPSCC). In the following sections, we will discuss current clinical evidence and the trials being conducted. 

### 5.1. Concurrent Immunotherapy with Radiotherapy in Locally Advanced HNSCC

Patients with locally advanced HNSCC have a 5-year OS of only 50% with the current standard treatment of concurrent chemoradiotherapy [2]. Although patients with HPV-associated oropharyngeal SCC have better OS, the 3-year OS is still around 70% for those with high-risk disease [21]. Strategies to improve survival with intensified therapy for patients with high-risk disease have been limited by treatment tolerability due to toxicities [29,55,56]. For those who are fit for cisplatin chemotherapy, a phase IB prospective trial demonstrated that pembrolizumab (200 mg intravenously 7 days before chemoradiotherapy, 2 additional dose on days 15 and 35 during chemoradiation) can be safely delivered with weekly cisplatin 40 mg/m^2^ and concomitant radiation at 70 Gy [57], with acute toxicities such as mucositis, radiation dermatitis, and dysphagia limited to grade 3. Despite a short follow-up, the early efficacy data was encouraging with 97.1% OS at 2 years in the HPV-associated group (*n* = 34). Only 1 distant failure occurred. For the HPV-negative cohort (*n* = 23), the follow up was too short to draw any conclusions. Disappointingly, a phase III JAVELIN 100 trial [58] (Table 1) that added concurrent and adjuvant avelumab to high dose cisplatin (100 mg/m^2^ on weeks 1, 5 and 7) with 70 Gy RT was terminated in 2019 after the planned interim analysis indicated that adding avelumab to cisplatin with concurrent RT did not demonstrate a statistically significant improvement in PFS [59]. The other two large phase III trials—Keynote 412 and REACH I (Table 1)—have completed recruitment in 2019, and the results are eagerly awaited. The feasibility and safety assessment of combining nivolumab and ipilimumab, a CTLA-4 inhibitor, with concurrent RT in patients with high risk LA-HNSCC was assessed in a single institution clinical trial (*n* = 24) [60], and early results showed 22% soft tissue ulceration at 3 months post-RT. The management of these ulcerations can be challenging including hyperbaric oxygen, lingual artery embolization and surgical debridement. Radiation-related osteonecrosis, and persistent inflammation were also reported in four patients amongst a small cohort. The RTOG 3504 trial included nivolumab before and after radiotherapy with a variety of concomitant systemic drugs. Whilst it was possible to safely combine nivolumab with all regimens, adjuvant nivolumab after radiotherapy and high dose cisplatin led to excessive Grade ≥ 3 immune related side effects but was feasible in those treated with concurrent weekly cisplatin or cetuximab [61]. Overall, caution is required with treatment escalation when incorporating ICIs into the treatment. Further data is required to support the approach of adding ICIs to standard cisplatin-based chemoradiotherapy in LA-HNSCC.

For those who are not eligible for cisplatin chemotherapy, concurrent cetuximab is the most common concurrent systemic therapy agent for LA-HNSCC treatment. ICI may potential be an agent to improve treatment outcomes in this population. The combination of conventional cetuximab-RT with avelumab (concurrent 10 mg/kg every 2 weeks followed by 4-month maintenance) has been shown to be feasible in a phase I trial [62]. This provides a ground for REACH II (Table 1) to evaluate adding concurrent and maintenance avelumab to concurrent cetuximab-RT. The safety cohort of the first 82 patients randomized showed no increase in grade 3 toxicity, and the trial has been approved to continue the trial without modifications [62]. The PembroRad, a phase II trial (Table 1) to compare concurrent pembrolizumab with cetuximab in concomitance with RT was recently presented at ESMO 2020. Concomitant pembrolizumab with RT did not improve cancer outcomes but appeared less toxic [15]. A phase II/III NRG-HN004 (Table 1) trial is still at phase II stage. It compares the durvalumab, a PD-L1 antibody, with cetuximab and the primary endpoint being OS rate.

### 5.2. Adjuvant Immunotherapy Post Curative Intent (Chemo) Radiotherapy in HNSCC

In the landmark PACIFIC trial, adjuvant therapy with durvalumab was given following radical chemoradiotherapy in patients with stage III NSCLC, showing significantly improved median PFS (17.2 vs. 5.6 months) and OS at 24 months (66.3%, CI 61.7–70.4 vs. 55.6%, CI 48.9–61.8, *p* = 0.005) [63]. This strategy is currently being investigated in the IMvoke010 trial for to test the adjuvant monotherapy with atezolizumab in LA-HNSCC patients post curative chemoradiotherapy (Table 1), with the aim to improve local and distant disease control. In the experimental arm, patients receive atezolizumab 1200 mg every 3 weeks for up to 1 year. The primary endpoints are investigator-assessed event-free survival and OS at 54 months after randomization. This study is currently recruiting. 

Since 2015, the combined RTOG 9501/Intergroup [64] and EORTC 22,931 [65] analysis identified extracapsular extension and positive margins as high-risk features for HNSCC disease recurrence post-surgery. It has been well demonstrated that concurrent cisplatin with radiotherapy (66 Gy) improves OS in this population [66]. More recently, the potential benefit from the addition of immunotherapy to chemoradiotherapy is being investigated in the GORTEC 2018-01 NIVOPOSTOP phase III randomized trial (Table 1).

### 5.3. Neo-Adjuvant Immunotherapy 

As discussed above, patients with operable LA-HNSCC requires intensive post-operative cisplatin and radiotherapy, however, 35% of patients, particularly those with HPV-negative HNSCC, will develop disease relapse [64,65]. The administration of ICIs prior to the surgery can potentially reduce the risk of subsequent disease recurrence post-operatively and downstage the tumor pre-operatively. Induction chemotherapy has not been proven to improve overall survival when added to concurrent chemoradiotherapy in LA-HNSCC [55,67]. Phase 2 studies (Table 1) with neoadjuvant and adjuvant pembrolizumab have demonstrated pathological responses in approximately 40% of the patients, with acceptable safety profile [68]. The randomized phase III Keynote-689 trial (Table 1), which is currently recruiting, will evaluate the efficacy and safety of neoadjuvant pembrolizumab and adjuvant pembrolizumab plus standard of care in patients with previously untreated resectable LA HNSCC. 

### 5.4. HPV-Associated Oropharyngeal Squamous Cell Carcinoma (HPV-OPSCC)

Patients with HPV-OPSCC have favorable disease control rates and overall survival rates compared to their HPV-negative counterparts [21,22,23]. HPV-OPSCC has a different immunophenotype from HPV-negative HNSCC with increased T-cell infiltration, natural killer cells recruitment, PD-L1 and CTLA-4 expression, and higher tumor mutational burden [69] compared to HPV-negative tumors. In recent years, there has been an increasing interest to explore ICIs as a treatment de-escalation approach in the treatment of HPV-OPSCC in recent years. KEYCHAIN (NCT03383094) and CITHARE (NCT03623646) are two phase II trials investigating pembrolizumab and duvalumab, respectively, in place of cisplatin with radiotherapy to 70 Gy (Table 2). The NRG-HN005 (NCT03952585) is a phase II/III randomized trial comparing nivolumab to cisplatin, with concurrent radiotherapy in patients with low-risk HPV-OPSCC (T1-2 N1 M0 or T3 N0-1 M0 AJCC 8th edition with smoking history <10 pack-years). This trial also compares the two arms with reduced radiotherapy dose to standard dose radiotherapy (70 Gy). The trial is currently in phase II stage to demonstrate non-inferiority in terms of PFS of concurrent reduced-dose RT (60 Gy) with nivolumab or concurrent reduced-dose RT with cisplatin, which is the winning arm in the NRG-HN002 trial [70], to the concurrent standard-dose RT (70 Gy) with high dose cisplatin. In the phase III stage, the trial aimed to demonstrate co-primary endpoints of non-inferiority in PFS and superiority in quality of life as measured by the MDADI of concurrent reduced-dose radiation with nivolumab or cisplatin (Table 2).

A recent study has shown that high intratumoral immune cell (ITIC) CD103 expression (>30%), a marker of tissue-resident memory T cells, is associated with better prognosis in patients with HPV OPSCC [71]. This was further confirmed in the pooled analysis of TROG 12.01 and De-ESCALaTE trials, where ITIC CD103 expression can separate low-risk HPV-OPSCC patients treated with cetuximab-RT into subgroups of excellent and poor prognoses [72]. Populations with high ITIC CD103 expression are a potential target for future de-escalation trials.

## 6. Recurrent or Metastatic HNSCC and Stereotactic Body Radiotherapy (SBRT)

Since Hellman and Weichselbaum [73] first proposed the existence of a clinically significant state of *oligometastases* (i.e., tumor metastases to a single or a limited number of organs prior to widespread dissemination), it poses an attractive idea to cure these patients with aggressive local treatment. It has been reported that undergoing salvage surgery and the presence of oligometastatic disease were associated with better OS in recurrent or metastatic HNSCC [74,75]. With the development of stereotactic body radiotherapy (SBRT) technique, which allows precise delivery of an ablative dose of radiotherapy in a small number of fractions (typically 1–10), it has gained significant momentum in recent years to use SBRT to treat oligometastatic disease. To date, there are five reported phase II trials to support the clinical benefit of SBRT in terms of PFS or even OS for patients with oligometastatic disease especially in non-small cell lung cancer [76,77] and prostate cancer [78,79]. The phase II SABR-COMET trial, which included a small number of patients with HNSCC, demonstrated that comprehensive local ablation of all metastatic disease sites (1–5 metastases) has an OS benefit (42% vs. 17% at 5 years, *p* = 0.006) compared with conventional palliative treatments [80]. The role of radiotherapy in metastatic tumors has expanded to include stereotaxic ablative radiotherapy for oligometastases, as well as more conventional symptom palliation. There is a growing body of evidence, albeit retrospective, to support the use of SBRT for patients with oligometastatic HNSCC [81,82,83,84,85], however, prospective randomized trials are yet to be conducted.

## 7. Combing Radiotherapy and Immunotherapy in Recurrent or Metastatic HNSCC: Abscopal Effect in HNSCC

The systemic effect of radiotherapy, or “abscopal effect” (from the Latin *ab scopus*, away from the target) was first introduced by Mole in 1953 to describe the phenomenon of regression of metastatic cancer “at a distance from the irradiated volume but within the same organism” [86]. It has been generally accepted that immune mechanisms are the underlying driving forces [87,88]. With increasing utilization of immunotherapy [89], abscopal effects are being increasingly observed, and efforts to exploit this effect are increasing [90]. Pre-clinical studies have demonstrated synergistic effects of combining immune therapy with radiotherapy to induce the “abscopal effect” [46,91,92]. Since Postow et al. [93] reported a sustained complete response in a patient with metastatic melanoma after treatment with a combination of SBRT to a single paraspinal mass and ipilimumab, the abscopal response has been recognized in an increasing number of case reports of patients receiving radiotherapy and immunotherapy in various tumor types including HNSCC [94,95]. Many patients are appropriate candidates for both radiotherapy and immunotherapy, highlighting the need for data to guide this treatment combination in patients with metastatic disease. The safety of the combined treatment and the impact of the dose, timing, and site of radiotherapy are important factors in the management of the cancer patient [96]. Multiple phase I/II studies that investigate the treatment combination’s efficacy and evaluate the safety are ongoing. In the studies published thus far, it appears that the combination of SBRT and ICIs is tolerable [14]. In a recent pooled analysis of phase 2 PEMBRO-RT trial (NCT02492568) and the MD Anderson trial (NCT02444741), Theelen et al. [97] reported that the concurrent pembrolizumab plus SBRT to one metastatic site leads to 41.7% out-of-field (abscopal) response in patients with metastatic non-small cell lung cancer. The disease control rate in the combined pembrolizumab and SBRT arm was superior compared to the arm of pembrolizumab alone (65.3% vs. 43.4%, *p* = 0.0071). Patients in the SBRT arm also had an improvement in PFS (9.0 vs. 4.4 months) and OS (19.2 vs. 9 months) compared with those who had pembrolizumab alone. However, in a recent phase 2 study that combine nivolumab with SBRT in metastatic HNSCC, there was no significant difference in objective response rate when SBRT was added to nivolumab compared with nivolumab alone [98]. Interestingly, patients with HPV-OPSCC had an inferior response than HPV negative OPSCC when they are treated with SABR with nivolumab. Additional investigation is warranted to determine the optimal radiotherapy dose and timing, immunotherapeutic agents, number of lesions to be treated, and the appropriate patient cohorts to fully evaluate the potential of the combination of SBRT and immunotherapy in metastatic HNSCC. Currently, a phase I/II trial (NCT03283605) is recruiting to assess the safety and efficacy of a triple treatment combination consisting of the administration of duvalumab and trimelimumab in combination with SBRT to 2–5 extracranial metastatic lesions in HNSCC [16]. 

## 8. Conclusions

The HNSCC immune tumor microenvironment is diverse and heterogeneous [69]. The therapeutic potential of radioimmunotherapy in HNSCC is highly promising and the field awaits results from ongoing clinical trials. Although the HPV-related OPSCC cases have more favorable outcomes compared to their HPV-negative counterparts, recent phase 3 studies comparing cisplatin to cetuximab (drug that was expected to have less toxicity) showed that the latter had poorer survival outcomes and disease control than the cisplatin arm, with no difference in toxicities [35,36]. These results serve as a reminder that treatment de-escalation requires careful consideration to achieve the fine balance of cure versus treatment toxicity, particularly in the growing population of young, HPV-related HNSCC patients. To date, no clinical trial has shown that tumoral HPV status predicts the treatment response to immunotherapy in metastatic disease [11], after controlling for PD-L1 status. An accurate prognostic stratification of HPV-associated oropharyngeal carcinoma is required to identify patients who are good responders to immunotherapy and are potentially suitable for treatment de-escalation. 

## Figures and Tables

**Table 1 cancers-13-05716-t001:** Phase II/III trials involving RT with immunotherapy in locally advanced HNSCC.

Study Name/NCT Number	Phase	Study Arm	Reference Arm	No.	Primary Endpoint	Status/Result
*Concurrent immunotherapy—eligible for high-dose cisplatin*						
JAVELIN 100 NCT02952586	III	Avelumab + Cis + RT + adjuvant Avelumab	Cis + Placebo + RT	697	PFS	Terminated in 2019 due to futility
REACH I NCT02999088	III	Avelumab + Cis + RT Avelumab + Cetuximab + RT	Cis +RT	688	PFS at 6 years	Accrual completed in 2019
Keynote 412 NCT03040999	III	Pembro + Cis + RT +adjuvant Pembro	Cis + Placebo + RT	780	EFS at 5 years	Accrual completed in 2019
CA2099TM Cohort 2 NTC03349710	III	Nivolumab +Cis + RT → adjuvant Nivo *6 cycle	RT + Cis → Placebo	68	Incidence of adverse events up to 12 months	Closed
CA2099TM Cohort 1	III	RT + Nivo → adjuvant Nivo *6 cycle	RT + Cetuximab → Placebo			
*Concurrent immunotherapy—ineligible for high-dose cisplatin*						
PembroRad GORTEC 2015-01 NCT02707588	II	RT + Pembro	RT + Cetuximab	131	Locoregional control at 15 months	Accrual completed in 2018
NRG-HN004 NCT03258554	II/III	RT + Durvalumab	RT + Cetuximab	523	Phase II: dose-limiting toxicity and PFS Phase III: OS	Ongoing (phase II)
REACH II GORTEC NCT02999087	III	RT +Avelumab + cetuximab + adjuvant avelumab	RT + Cetuximab	420	PFS	Active, not recruiting
*Neoadjuvant immunotherapy*						
NCT02641093	II	Pembro → surgery → adjuvant RT + Pembro +/− weekly cis	N/A	76	1-year DFS	1-year DFS 67% for high risk, 93% for intermediate risk; 32/76 pathological response
NCT02296684	II	Cohort 1: Pembro *1c → surgery → RT +/− cis + pembro *6c Cohort 2: Pembro *2c → surgery → RT +/− cis	N/A	67	Locoregional recurrence; distant failure rate and major pathologic response	44% pathological response
KEYNOTE-689 NCT03765918	III	Pembro *2 cycles → surgery → RT +/− cis + pembro *15c	Surgery → chemoRT	600	Pathological response EFS	Recruiting
IMSTAR-HN NCT03700905	III	Nivo → surgery → chemoRT + adjuvant nivolumab +/− ipi	Surgery → chemoRT	276	DFS	Active
CheckRad-CD8 NCT03426657	II	Durvalumab/cisplatin/docetaxel → re-biopsy, if 20% increase in CD8+ T cell or pCR → 70 Gy + durvalumab/tremelimumab *3 cycles → duvlalumab *8 cycles	N/A	80	Feasibility rate of patients entering radio-immunotherapy to receive treatment until at least cycle 6 of immunotherapy of ≥80%	Meet the primary feasibility endpoint
*Adjuvant immunotherapy*						
Invoke 010 NCT03452137	III	chemoRT → Atezolizumab	chemoRT → Placebo	400	EFS and OS	Recruiting
NIVOPOSTOP NCT03576417	III	Surgery → Nivolumab (240 mg) → Cis +RT (66 Gy) + Nivolumab (360 mg) → Nivolumab (480 mg)	Surgery → cis + RT (66 Gy)	680	DFS	Recruiting

Cis, cisplatin; Pembro, Pemrolizumab; Nivo, Nivolumab; Ipi, ipilimumab; Atezolizumab, Anti PD-L1; DFS, disease free survival; PFS, progression free survival; EFS, event free survival; pCR, pathological complete response.

**Table 2 cancers-13-05716-t002:** Clinical trials involving RT with immunotherapy in HPV related oropharyngeal carcinoma.

Study Name/NCT Number	Phase of Trial	Study Arm	Reference Arm	No.	Primary Endpoint	Status/Result
*Concurrent immunotherapy*						
NRG-HN005 NCT03952585	II/III	Arm 1: RT (60 Gy in 5 weeks using 6 fractions per week) + Nivolumab * 6c Arm 2: RT (60 Gy in 6 weeks using 5 fractions per week) + Cis (days 1 and 22)	Arm3: RT (70 Gy in 6 weeks using 6 fractions per week) + Cis (days 1 and 22)	711	PFS and quality of life	Recruiting
KEYCHAIN NCT03383094	II	RT (70 Gy over 6.5 weeks) + Pembro (200 mg every 3 weeks *20 cycles)	RT (70 Gy over 6.5 weeks) + Cis (100 mg/m^2^ weeks 1, 4 and 7)	114	PFS at 3 years	Recruiting
CITHARE NCT03623646	II	RT (70 Gy) + Durvalumab	RT (70 Gy) + Cis	66	PFS at 12 months	Active, not recruiting
*Neoadjuvant immunotherapy*						
OPTIMA II NCT03107182	II	Nivo/nab-paclitaxel/carboplatin induction → adaptive treatment as per response to induction treatment	Cis-RT	73	Deep response rate ≥50% shrinkage to induction	70.8% pts have tumor shrinkage >50%;
NCT03618134	I/II	SBRT + Durvalumab +/− tremelimumab → TORS → Duvalumab	NA	82	Phase I: Incidence of adverse events Phase II: PFS	Recruiting

Cis, cisplatin; Pembro, Pemrolizumab; Nivo, Nivolumab; Ipi, ipilimumab; PFS, progression free survival; SBRT, sterotactic body radiotherapy; TORS, transoral robotic surgery.

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
