# Peer review of "Recent Research on Combination of Radiotherapy with Targeted Therapy or Immunotherapy in Head and Neck Squamous Cell Carcinoma: A Review for Radiation Oncologists"

_cancers, 2021, doi:10.3390/cancers13225716_

Round 1

Reviewer 1 Report

The review paper Xing et al. discuss the latest evidence to support combinations of RT with targeted therapies (cetuximab) or immunotherapy (ICIs) compared to standard of care treatment (RT+cisplatin) in HNSCC. The authors summarize the findings from recent clinical trials that incorporate ICI with RT and discuss the potential of combining ICI and RT as a treatment de-escalating strategy in HPV-related OPSCC.

The paper is well written, with a comprehensible representation of current evidence on combining targeted therapies and immunotherapy with RT in HNSCC. However, as the review is clearly focused in representation of clinical trial data, I would suggest to remove or extensively shorten subsections 3. and 4. in which biological background of tumor microenvironment and mechanisms of immunological cell death after RT are presented, as these data do not follow the clinical flow of the paper.

In subsection 6. the relation of SBRT with targeted therapies or immunotherapy in treatment of oligometastatic disese is also not clear, as this is the primary focus of the review.

In subsection 5.1 the word data is repeated in the sentence twice.

Author Response

The review paper Xing et al. discuss the latest evidence to support combinations of RT with targeted therapies (cetuximab) or immunotherapy (ICIs) compared to standard of care treatment (RT+cisplatin) in HNSCC. The authors summarize the findings from recent clinical trials that incorporate ICI with RT and discuss the potential of combining ICI and RT as a treatment de-escalating strategy in HPV-related OPSCC.

The paper is well written, with a comprehensible representation of current evidence on combining targeted therapies and immunotherapy with RT in HNSCC. However, as the review is clearly focused in representation of clinical trial data, I would suggest to remove or extensively shorten subsections 3. and 4. in which biological background of tumor microenvironment and mechanisms of immunological cell death after RT are presented, as these data do not follow the clinical flow of the paper.

Response: Thank you for your suggestion. In sections 4., we very briefly summarized the potential mechanism of radiation induced immune response and provide a background to the rationale and need for clinical studies evaluating combination of radiation and immunotherapy and targeted therapies. We feel it is helpful for the readers with limited knowledge of radiotherapy to better understand the rationale of combination of radiation therapy with immunotherapy in cancer management. Subsections 3 and 4 provides a summary of studies to date before subsection 6. These are current evidence generated to date to justify for clinical studies/ interventional studies.

In subsection 6. the relation of SBRT with targeted therapies or immunotherapy in treatment of oligometastatic disease is also not clear, as this is the primary focus of the review.

Response: We further discuss the relationship of SBRT with immunotherapy in section 7. We have used subsection 6 to introduce the concept of SBRT in head and neck cancer, and subsection 7 to demonstrate current evidence for combination of SBRT and immunotherapy.

In subsection 5.1 the word data is repeated in the sentence twice.

Response: the additional word data is now deleted.

Reviewer 2 Report

This manuscript entitled Recent Research on Combination of Radiotherapy with Targeted Therapy or Immunotherapy in Head and Neck Squamous Cell Carcinoma: A Review for Radiation Oncologists review the  relevant literature and the status of current ongoing clinical trials and summarise several important results in this field. This manuscript is well organized and comprehensively described.

One minor error:

At the cover page Abstract line 10 "Cetuximab is suitable for patients who are not fit for radiotherapy" should be corrected as "Cetuximab is suitable for patients who are not fit for cisplatin". 

Author Response

This manuscript entitled Recent Research on Combination of Radiotherapy with Targeted Therapy or Immunotherapy in Head and Neck Squamous Cell Carcinoma: A Review for Radiation Oncologists review the relevant literature and the status of current ongoing clinical trials and summarise several important results in this field. This manuscript is well organized and comprehensively described.

One minor error:

At the cover page Abstract line 10 "Cetuximab is suitable for patients who are not fit for radiotherapy" should be corrected as "Cetuximab is suitable for patients who are not fit for cisplatin". 

Response 2: This is now corrected.

Reviewer 3 Report

Interesting paper. To improve the overall quality following suggestions:

Introduction

  • Transoral surgery, through increasingly less invasive approaches, reduced bleeding rates, and reduced dysphagia even for carcinomas of the oropharynx provides a therapeutic approach that in conjunction with chemoradiotherapy can provide a therapeutic option. please cite doi:10.1002/lary.27567 and doi:10.1016/j.anl.2021.05.007
  • In patients with T1-2, N3 HNSCC undergoing upfront neckdissection, radiotherapy and surgery of the primary produce similar oncological outcomes. Morbidity was related to the extent of the neck dissection, thus proposing a comparable therapeutic option depending on the clinical extent detected. please cite doi:10.1245/s10434-019-07589-0

Section 5

  • Immune checkpoint inhibitors have emerged as a breakthrough therapy in the treatment of various metastatic cancers. In parallel, the role of radiotherapy in metastatic tumors has expanded to include stereotaxic ablative radiotherapy for oligometastases, as well as more conventional symptom palliation. Therefore, many patients are appropriate candidates for both radiotherapy and immunotherapy, highlighting the need for data to guide this treatment combination in patients with metastatic disease. The safety of the combined treatment and the impact of the dose, timing and site of radiotherapy are important factors in the management of the cancer patient. please cite doi:10.21037/apm.2018.07.10

Section 6

  • A recent randomized Phase II Trial analyzes the role of Nivolumab With Stereotactic Body Radiotherapy Versus Nivolumab Alone in Metastatic Head and Neck Squamous Cell Carcinoma. The authors found no improvement in response and no evidence of an abscopal effect with the addition of SBRT to nivolumab in unselected patients with metastatic HNSCC. please discuss and cite doi:10.1200/JCO.20.00290

Author Response

Interesting paper. To improve the overall quality following suggestions:

Introduction

  • Transoral surgery, through increasingly less invasive approaches, reduced bleeding rates, and reduced dysphagia even for carcinomas of the oropharynx provides a therapeutic approach that in conjunction with chemoradiotherapy can provide a therapeutic option. please cite doi:10.1002/lary.27567 and doi:10.1016/j.anl.2021.05.007
  • In patients with T1-2, N3 HNSCC undergoing upfront neck dissection, radiotherapy and surgery of the primary produce similar oncological outcomes. Morbidity was related to the extent of the neck dissection, thus proposing a comparable therapeutic option depending on the clinical extent detected. please cite doi:10.1245/s10434-019-07589-0

Response: Thank you for your suggestion. We agree that transoral surgery is another good treatment option in managing oropharynx and oral cavity cancers. However, it is beyond the scope of this review which is to summarise the current evidence for radiotherapy and targeted therapies/ immunotherapies. To include transoral surgery in the introduction but no further mention in the manuscript may confuse readers as to the aim of this review.

Section 5

  • Immune checkpoint inhibitors have emerged as a breakthrough therapy in the treatment of various metastatic cancers. In parallel, the role of radiotherapy in metastatic tumors has expanded to include stereotaxic ablative radiotherapy for oligometastases, as well as more conventional symptom palliation. Therefore, many patients are appropriate candidates for both radiotherapy and immunotherapy, highlighting the need for data to guide this treatment combination in patients with metastatic disease. The safety of the combined treatment and the impact of the dose, timing and site of radiotherapy are important factors in the management of the cancer patient. please cite doi:10.21037/apm.2018.07.10

Response: Thank you for your suggestions. We have now added it to Section 6. and Section 7.

Section 6

  • A recent randomized Phase II Trial analyzes the role of Nivolumab With Stereotactic Body Radiotherapy Versus Nivolumab Alone in Metastatic Head and Neck Squamous Cell Carcinoma. The authors found no improvement in response and no evidence of an abscopal effect with the addition of SBRT to nivolumab in unselected patients with metastatic HNSCC. please discuss and cite doi:10.1200/JCO.20.00290

Response: This study is now discussed and cited in section 7.